# Antimicrobial Activity of 1,3,4-Oxadiazole Derivatives

**DOI:** 10.3390/ijms22136979

**Published:** 2021-06-29

**Authors:** Teresa Glomb, Piotr Świątek

**Affiliations:** Department of Medicinal Chemistry, Faculty of Pharmacy, Wroclaw Medical University, Borowska 211, 50-556 Wroclaw, Poland; teresa.glomb@umed.wroc.pl

**Keywords:** 1,3,4-oxadiazole, antimicrobial activity, antibacterial activity, antifungal activity, antiprotozoal activity, antiviral activity

## Abstract

The worldwide development of antimicrobial resistance forces scientists to search for new compounds to which microbes would be sensitive. Many new structures contain the 1,3,4-oxadiazole ring, which have shown various antimicrobial activity, e.g., antibacterial, antitubercular, antifungal, antiprotozoal and antiviral. In many publications, the activity of new compounds exceeds the activity of already known antibiotics and other antimicrobial agents, so their potential as new drugs is very promising. The review of active antimicrobial 1,3,4-oxadiazole derivatives is based on the literature from 2015 to 2021.

## 1. Introduction

Antimicrobial resistance (AMR) is one of the main problems of modern medicine. Poor treatment of infections, over-prescription of antibiotics and their inappropriate use by patients have made some of the microorganisms insensitive to currently used drugs. This causes great difficulties in treatment as the antibiotics or other antimicrobial drugs used so far are no longer effective and infections become progressively difficult to treat [1,2].

AMR is an increasingly serious threat to life and public health. Without effective antibiotic therapy, the cost of caring for patients with drug-resistant infections increases, and there is a huge risk during surgery and other medical procedures [3,4]. Antimicrobial resistance occurs when microorganisms develop the ability to defeat drugs designed to kill them. There is great diversity of microbial defense strategies [1].

One of the ways to deal with the AMR problem is the synthesis of new medicinal substances to which microorganisms are sensitive. Researchers around the world are working on new molecules that would stop the development of resistance. Most often, newly formed compounds contain a heterocyclic moiety, and the motifs bearing the 1,3,4-oxadiazole ring constitute a large group of potential antimicrobial derivatives.

Oxadiazoles are five-membered heterocyclic compounds containing two nitrogen atoms and one oxygen atom in their structure. We can distinguish several isomeric forms of oxadiazole, which occur in the structure of many drugs, e.g., anticancer zibotentan [5], antimicrobial furamizole [6], antiviral raltegravir [7], ataluren for Duchenne muscular dystrophy [8] and others (Figure 1).

The most promising structures are 1,3,4-oxadiazole derivatives. At the end of the nineteenth century, the first derivatives of 1,3,4-oxadiazole were synthesized. The methods of obtaining the new structures were multidirectional, including reactions of appropriate hydrazides and phosgene [9], thermal cyclization of 1-acylsemicarbazides [10] or cyclization of 1,2-diacylhydrazines by the action of dehydrating agents [11]. The amount of research on this molecule was intensified in the 1950s through the 1960s of the twentieth century [12,13]. Currently, scientists use various routes for the preparation of 1,3,4-oxadiazole derivatives, some of them are improved previous methods, e.g., cyclization oxydative reactions of *N*-acylhydrazones, cyclodehydration reactions of diacylhydrazines or hydrazide reactions with carbon disulfide [14]. In the last two decades, a significant increase in research on the 1,3,4-oxadiazole ring can be observed [15].

The presence of the 1,3,4-oxadiazole ring affects the physicochemical and pharmacokinetic properties of the entire compound. The 1,3,4-oxadiazole ring has aroused interest in medicinal chemistry as a bioisostere for carbonyl containing molecules such as carboxylic acids, esters and amides. The oxadiazole ring is used also as a significant part of the pharmacophore which is capable of binding to a ligand. In some cases, it acts as a flat aromatic linker to ensure proper orientation of the structure [16].

These features of the 1,3,4-oxadiazole ring have resulted in a wide variety of pharmaceutical applications of these molecule. According to the literature, for many years, scientists around the world have been synthesizing new compounds containing the 1,3,4-oxadiazole core, which showed a wide spectrum of biological activity, including anti-inflammatory [17,18,19,20], analgesic [21], anti-depressive [22,23], anticancer [24,25] and anti-diabetic [26,27] effect. Moreover, there are numerous literature reports confirming the broad antimicrobial activity of compounds containing the 1,3,4-oxadiazole ring in its structure. Scientists have conducted research on antibacterial [28,29,30], antifungal [31,32] or antiviral [33,34] molecules bearing a 1,3,4-oxadiazole scaffold.

Due to the extensive antimicrobial activity of 1,3,4-oxadiazole derivatives, in this review we want to summarize the achievements of scientists over the last seven years. This review is based on articles from 2015 to 2021; the PubMed database was used to search for literature, entering the keywords “1,3,4-oxadiazole”, “antibacterial”, “antitubercular”, “antifungal”, “antiprotozoal”, “antiviral”, and “activity”. Structures were divided according to individual antimicrobial activities; where possible, a division based on chemical structure was also used.

## 2. Antibacterial Activity of 1,3,4-Oxadiazole Derivatives

### 2.1. 1,3,4-Oxadiazole Hybrids of Quinolone Antibacterial Drugs

Known quinolone antibacterial drugs have become the basis for scientists to modify their structure.

For Peraman et al. (2015), nalidixic acid was a pharmacophore moiety for structure modification. The carboxylic group was replaced with a thiosemicarbazide/acidcarbazide chain or thioxo-1,2,4-triazole/1,3,4-oxadiazole linked with quinoxaline. The compound containing 1,3,4-oxadiazole (**1**, Figure 2) showed stronger or comparable activity against *Pseudomonas aeruginosa* and *Staphylococcus aureus* than the reference drugs (ciprofloxacin and amoxicillin). Additionally, the new derivatives showed antitubercular activity [35].

Moreover, Omar et al. (2018) synthesized nalidixic acid derivatives replacing the carboxylic group with the isosteric 1,3,4-oxadiazole ring. The activity against selected bacterial Gram-positive strains *S. aureus*, *Bacillus cereus* and Gram-negative strains *Escherichia coli*, *Klebsiella pneumoniae*, *P. aeruginosa* of the most active derivatives **2a**–**b**, **3a**–**b** (Figure 2) was even 2–3 times stronger than the reference nalidixic acid. New compounds were also tested for inhibition of DNA gyrase as a potential mechanism of action [36]. DNA gyrase, otherwise known as topoisomerase II, is essential for chain elongation during replication of the chromosome in bacterial cells [37].

In 2019, a team of researchers led by Guo developed an innovative series of norfloxacin derivatives containing a 1,3,4-oxadiazole ring. The most potent compounds **4a**–**c** (Figure 3) showed excellent antibacterial activity against *S. aureus* (minimum inhibitory concentrations-MICs = 1–2 µg/mL) and methicillin-resistant *S. aureus* (MRSA1-3) strains (MICs = 0.25–1 µg/mL) compared to the reference drugs: norfloxacin and vancomycin (MICs = 1–2 µg/mL). The **4a** derivative also showed very good time-kill kinetics in relation to vancomycin [38].

In addition, Mermer et al. (2019) designed and synthesized hybrid fluoroquinolone-piperazine-azole (triazoles or 1,3,4-oxadiazoles) compounds. New derivatives were tested for their antibacterial activity against Gram-positive and Gram-negative bacteria. 1,3,4-oxadiazole derivatives **5a** and **5b** (Figure 3) showed good-to-excellent activity compared to the reference antibiotics: ampicillin and gentamicin. In addition, studies of new compounds as DNA gyrase inhibitors were carried out, trying to explain the mechanism of action of the new derivatives [39].

When analyzing the data presented above, it can be seen that modifications of quinolone antibacterial drugs are conducted in two directions. In the case of nalidixic acid, the 1,3,4-oxadiazole ring replaces the carboxylic moiety as a bioisosteric structure. In the case of fluoroquinolones (norflaxacin/ciprofloxacin), the 1,3,4-oxadiazole molecule is introduced at the piperazine substituent preceded by a methylene linker. Methoxy or halogen substituents on the aromatic ring often enhance the activity of the derivatives.

### 2.2. Antibacterial Activity of Aryl/Heteroaryl Derivatives of 1,3,4-Oxadiazole

Compounds with antibacterial activity are also sought among aromatic or heteroaromatic 1,3,4-oxadiazole derivatives. Dividing the new compounds due to the direct surrounding of the 1,3,4-oxadiazole ring, we can distinguish aryl derivatives, aryl and/or heteroaryl structures and derivatives with an aryl and/or heteroaryl bi-/tricyclic ring.

Antimicrobial activity is demonstrated by structures with a 1,3,4-oxadiazole ring containing an aryl substituent direct to the heterocycle.

In 2016, Navin et al. developed a series of benzothiazepine and benzodiazepine derivatives of aryl-1,3,4-oxadiazole and tested them for broad antimicrobial activity. The most active derivatives shown as a general structure **6** in Figure 4 turned out to be stronger than the reference drug ampicillin against *P. aeruginosa* and *S. aureus* strains. Additionally, antitubecular and antiprotozoal activities of the obtained compounds were demonstrated [40].

Alghamdi et al. from Saudi Arabia (2020) received a series of aryl-1,3,4-oxadiazole-benzothiazole derivatives, which they tested for antibacterial activity. The best compound **7** (Figure 4) having a thiol moiety displayed similar activity compared to the reference drug amoxicillin against Gram-positive bacteria (*Staphylococcus epidermidis*, *Staphylococcus aureus*, *Enterococcus faecalis*) [41].

1,3,4-oxadiazole derivatives directly containing an aryl and/or a heteroaryl (five or six membered ring) substituent also have antibacterial activity.

Dhumal et al. (2016) conducted research on an innovative combination of three heterocyclic rings: 1,3,4-oxadiazole, thiazole and pyridine and checked their antitubercular activity. The most active compounds were **8a** and **8b** (Figure 5), which strongly inhibited *Mycobacterium bovis* BCG both in the active and dormant state. Additionally, molecular docking studies were performed and the compounds’ binding affinity to the active site of mycobacterial enoyl reductase (InhA) enzyme was checked. This enzyme is a key regulatory factor in the biosynthesis of fatty acids. Inhibition of this enzyme may disrupt the synthesis of mycolic acid and, consequently, induce cell lysis [42].

In addition, Desai et al. (2016, 2018) conducted studies on the antitubercular effects of pyridine based 1,3,4-oxadiazole scaffold derivatives. The most active compounds of the series **9a**, **9b**, additionally containing the indole ring (Figure 5), showed strong activity against *M. bovis* BCG in the active and dormant state. In turn, benzhydrazide compounds **10a**–**c** (Figure 5) were more potent against the *M. tuberculosis* H37Ra strain. However, compared to the reference drugs isoniazid and/or rifampicin, the activity was much weaker. Additionally, active derivatives showed great potential during molecular docking studies with mycobacterial enoyl reductase (InhA) [43,44].

Moreover, in 2018, Mansoori et al. obtained acetylated nicotinic acid derivatives where carboxylic group was replaced with a 1,3,4-oxadiazole ring. The most active compounds had a nitro substituent in the phenyl ring **11a**–**c** (Figure 6) (the para position was preferred over meta and ortho) and showed tuberculostatic activity. Additionally, their antibacterial and antifungal properties were proven. The antimycobacterial activity was confirmed by molecular docking studies with appropriate proteins [45].

Another Indian scientists led by Ramesh M. Shingare (2018) obtained hybrid derivatives by combining 1,3,4-oxadiazole and isoxazole rings. The most active derivatives **12a**–**e** (Figure 6) showed antimicrobial activity 2–4 times stronger than the reference ampicillin against Gram-positive bacteria: *S. aureus*, *S. pyogenes* and Gram-negative: *P. aeruginosa* and *E. coli*. In contrast, compounds **12f** and **12g** (Figure 6) showed antimycobacterial activity against the *M. tuberculosis* H37Rv strain. Additionally, molecular docking studies were conducted to gain insight inhibition of the MurD ligase enzyme which is involved in the biosynthesis of cytoplasmic peptidoglycan precursor [46].

Das et al. (2020) synthesized a series of pyrazine containing 1,3,4-oxadiazoles with substituted azetidin-2-one. The compounds **13a** and **13b** (Figure 6) proved to be the most potent in terms of antibacterial activity, and showed moderate to excellent activity against *B. subtilis*, *S. aureus*, *E. coli*, and *P. aeruginosa* compared to the amoxicillin standard. In addition, these compounds have potent antifungal and antitubercular effect [47].

Also among heteroaryl bi- or tricyclic rings fused directly to 1,3,4-oxadiazole, compounds with antibacterial activity can be found.

Researchers from India under the direction of Sindhe (2016) synthesized 2,5-disubstituted 1,3,4-oxadiazole derivatives containing a naphthofuran moiety. A series of compounds were tested for antibacterial and antioxidant activity. The most active structures **14a**, **14b** (Figure 7) compared to ciprofloxacin, showed the same antibacterial effect (MIC = 0.2 mg/mL) against *P. aeruginosa* and *B. subtilis* and slightly weaker effect (MIC = 0.4 mg/mL) against *S. typhi* and *E. coli*. Additionally, they showed moderate antifungal activity [48].

In addition, Sajja et al. (2017) developed a series of 1,3,4-oxadiazole derivatives bearing a benzo[6.7]cyclohepta[1,2-*b*]pyridine moiety. The most active compound **15** (Figure 7) was twice as potent as the reference ethambutol against *M. tuberculosis* H37Rv, while it was less potent than isoniazid and rifampicin. Methoxyl substituents were key to enhancing antimycobacterial activity [49].

Moreover, Gholap et al. (2017) obtained a series of 1,3,4-oxadiazoles bearing 2,2-dimethyl-2,3-dihydrobenzofuran scaffold. The compounds were tested for tuberculostatic activity. The most potent compound **16** (Figure 8) showed activity against *M. tuberculosis* H37Ra ex vivo and in vitro, and against *M. bovis* BCG in vitro. Additionally, the molecular docking studies and testing of binding affinity to InhA proved the potential of new derivatives as antitubercular structures [50].

Furthermore, in 2018, Triloknadh et al. designed and synthesized a series of thieno[2,3-*d*]pyrimidine-1,3,4-oxadiazole hybrids and tested them for antimicrobial effect. The best activity was shown by compounds **17a**–**b** and **18a**–**b** (Figure 8) which additionally had morpholine or piperidine moiety in their structure. The derivatives acted similarly or strongly against the Gram-positive *S. aureus*, *B. subtilis* and Gram-negative strains *E. coli*, *P. aeruginosa* compared to gentamicin as the reference drug. Docking studies showed good binding affinity of the most active compounds to three selected bacterial target proteins. Moreover, some derivatives had neuroprotective activity [51].

Researchers from Poland under the direction of Paruch (2020) received a number of 2,5-distubstituted 3-acetyl-1,3,4-oxadiazole derivatives, which were tested for antibacterial activity. The most promising for further research or modification turned out to be compound **19** (Figure 8), which additionally contained a quinolin-4-yl substituent. It showed the strongest activity against *S. epidermidis* (MIC = 0.48 µg/mL) nearly 8 times higher than the reference nitrofurantoin [52].

Several conclusions regarding the structure-activity relationship can be drawn from the discussed derivatives. The type and position of the substituent on the aryl ring greatly influences the activity of the compounds; often the para position is preferred over other substitution sites. An additional heterocyclic ring connected to 1,3,4-oxadiazole enhances the antimicrobial effect. Antitubercular derivatives often have an additional pyridine ring in their structure, which is one of the key elements of the molecule.

### 2.3. Antibacterial Activity of Amino Derivatives of 1,3,4-Oxadiazole

Among many 1,3,4-oxadiazole derivatives, also the amino derivatives showed antibacterial activity.

Ladani et al. (2015) obtained a number of 2-amino-1,3,4-oxadiazole derivatives containing a quinoline ring in their structure. The new compounds were comprehensively tested for their antimicrobial activity. The strongest derivative **20** (Figure 9) bearing additionally pyridine moiety showed strong to moderate effect on strains of *C. tetani*, *B. subtilis*, *S. typhi*, and *E. coli* compared to ampicillin. In addition, it displayed good to excellent tubeculostatic and antimalarial activity [53].

Furthermore, Vosatka et al. from the Czech Republic (2018) received a series of 1,3,4-oxadiazole-2-amine motifs bearing a pyridine nucleus. The effect of the length of the alkyl substituent on the antitubercular activity of the molecule was studied in comparison to isoniazid. The most active compounds have 10 to 12 carbon atoms in the chain. Compound **21c** (Figure 9) *N*-dodecyl-5-(pyridin-4-yl)-1,3,4-oxadiazol-2-amine showed antimycobacterial activity on susceptible and drug-resistant *M. tuberculosis* strains (MICs = 4–8 µM). In turn, the *N*-decyl and *N*-undecyl derivatives **21a**–**b** (Figure 9) showed strong activity against the strains of *M. kansasii* 235/80 (MICs = 8–16 µM) [54].

Moreover, a new series of 2-acylamino-1,3,4-oxadiazole derivatives was developed by Li et al. (2019). Compound **22a** (Figure 9) appeared to be most active against *Staphylococcus aureus* (MIC = 1.56 µg/mL), while **22b** and **22c** (Figure 9) were most effective against *Bacillus subtilis* (MIC = 0.78 µg/mL); levofloxacin was used as a positive control [55].

In 2019, Hkiri et al. published the results of in vitro studies of the 2,5-diamino-1,3,4-oxadiazole derivatives, showing antibacterial activity against *Staphylococcus aureus*, *Enterococcus faecium*, *Streptococcus agalactiae*, *Escherichia coli* and *Salmonella typhimurium*. Their antimicrobial properties were determined on the basis of the inhibition zone diameter (IZD) using the filter paper disc-diffusion method. At the concentration of 15 µg/mL, most of the compounds showed good-to-excellent inhibitory activity against all bacterial strains, and the inhibition zone diameters ranged from 7 to 46.5 mm. Compound **23** (Figure 9) as symmetric bis-amidine 1,3,4-oxadiazole derivative, demonstrated the best antimicrobial activity, in some cases better than the control drug–ampicillin. Researchers considered the inhibition of enoyl-acyl carrier protein (ACP) reductase (FabI) as the probable mechanism of action, confirmed by the molecular docking analysis. It is an NADH(nicotinamide adenine dinucleotide)-dependent enzyme that takes part in the final stage of fatty acids synthesis. It catalyzes the reduction reaction to saturated acyl-ACP, which is an important step in the fatty acid elongation cycle [56].

In addition, Salama et al. (2020) synthesized a series of 2-amino-1,3,4-oxadiazole derivatives and checked their activity against the *Salmonella typhi* strain. The most active and promising derivatives turned out to be compounds **24**–**27** (Figure 10). Compounds **24a** and **24b** have simple structure, **25** is coupled with *N*-protected amino acids as *N*-Boc (*tert*-butyloxycarbonyl) phenylalanine. In turn the broadest spectrum activity among them was shown by compounds **26** and **27** which possess an additional heterocyclic ring (benzothiazole or thiazolidine) in their structure [57].

A team of researchers from Egypt led by Hagras, Hannoun and Kotb (2018–2020) designed and obtained several series of 5-(4-methyl-thiazol-5-yl)-1,3,4-oxadiazole-2-amine derivatives containing a *tert*-butylphenyl, biphenyl or naphthalene moiety. The series were prepared to improve the pharmacokinetic properties of the compounds and the antimicrobial activity. The most active derivatives of each series shown as general structure **28** in Figure 11 displayed activity against methicillin-resistant *Staphylococcus aureus* (2658 RCMB), which was stronger or comparable to vancomycin [58,59,60].

Scientists from the USA under the direction of Naclerio (2018–2021) have been verifying *N*-(1,3,4-oxadiazol-2-yl)-benzamide derivatives for their antibacterial activity. They optimalised the developed structures in order to strengthen their effect against specific strains. The most active derivatives bearing a (piperidin-1-yl)sulfonyl moiety turned out to be **29a** and **29b** (Figure 12), which showed 4 times more potent action against *S. aureus* and MRSA strains than vancomycin and was 8 times more active than linezolid. Compound **29b** was additionally twice as powerful as linezolid against vancomycin-resistant *Enterococcus* (VRE) strains and, as well as the reference medicine, it was active against *Enteroccocus faecalis* strains. Moreover, the derivatives’ mechanism of action as an inhibitor of the synthesis of lipoteicholic acid (LTA), which is essential for the growth, biofilm formation and virulence of many Gram-positive bacteria, was confirmed. Other benzamide derivatives had a pivotal effect on other strains. Compound **30** (Figure 12) showed significantly higher activity against different *Clostridium difficile* strains (MICs = 0.003–0.03 µg/mL) compared to the reference drug-vancomycin (MICs = 0.25–1 µg/mL). On the other hand, structures **31a** and **31b** (Figure 12) had much stronger effects on *Neisseria gonorrhoeae* strains (MICs = 0.03–0.125 μg/mL) compared to azithromycin or tetracycline (MICs = 0.25–4 μg/mL) [61,62,63,64,65].

In summary, several conclusions can be made regarding the structure-activity relationship. In the amino derivatives of 1,3,4-oxadiazole it can be seen that the additional presence of another heterocyclic ring can broaden the spectrum of antimicrobial activity. In the case of *N*-alkyl derivatives, the structures with 10 to 12 carbon atoms in the chain show the highest activity. In the symmetrically substituted 2,5-diamino-1,3,4-oxadiazole derivatives, the best activity had amidine structures, which were superior to amide or imine groups. In the 5-(4-methyl-thiazol-5-yl)-1,3,4-oxadiazole-2-amine derivatives, the improved anti-MRSA potency resulted from the increased polarity of the nitrogenous group by the introduction of hydrazine or guanidine moiety. On the other hand, taking into account benzamide derivatives, the trifluoromethylene group occurs in the most active compounds of each series (whether alone or in combination with an oxygen or sulfur atom). An additionally very important element for antibacterial activity is the presence of (3,5-dimethylpiperidin-1-yl)sulfonyl) moiety.

### 2.4. Antibacterial Activity of 1,3,4-Oxadiazole-2-Thiones/Thiols and their S-Substituted Derivatives

Antimicrobial activity was also characterized by 1,3,4-oxadiazole derivatives containing a free thione or thiol group or *S*-substituted structures. Among them, we distinguish simple or more complex molecules.

Karabanovich et al. (2016, 2017) conducted research on the antimycobacterial activity of S-substituted 1,3,4-oxadiazole-2-thiols derivatives. They received two series containing the 3,5-nitrophenyl moiety necessary for activity at the 5-position of 1,3,4-oxadiazole or linked through a thiol group. The most active derivatives of each series **32a**–**h**, **33a**–**d** (Figure 13) showed a much stronger effect (MIC = 0.03 µM) than the reference isoniazid (MIC = 0.5 µM) against *M. tuberculosis* My 331/88 and additionally displayed activity against nontuberculous strains of *M. kansasii* (MICs = 0.03–0.5 µM) [66,67].

In 2019, researchers under the direction of Makane synthesized a series of simple 5-substituted 2-mercapto-1,3,4-oxadiazole derivatives. Compound **34** (Figure 13) containing 4-hydroxyphenyl substituent in position 5 showed the strongest inhibitory activity against *M. tuberculosis* H37Rv and high selectivity. Moreover, it was also active against strains resistant to commonly used antitubercular drugs [68].

Moreover, Yarmohammadi et al. (2020) also obtained a number of simple 5-aryl-1,3,4-oxadiazole-2-thiol derivatives. The compounds were tested for antimicrobial activity. Compound **35** (5-(4-fluorophenyl)-1,3,4-oxadiazole-2-thiol) (Figure 13) had the greatest potential, as it showed stronger activity against *E.coli* and *S. pneumoniae* compared to ampicillin, and it was active against *P. aeruginosa*—over 100 times stronger. Additionally, it had antifungal activity against *A. fumigatus* better than terbinafine [69].

In turn, Aziz-Ur-Rehman and his team (2020) obtained 5-(3-chlorophenyl)-2-((*N*-(substituted)-2-acetamoyl)mercapto)-1,3,4-oxadiazole derivatives. The most active compound **36** (Figure 13) showed moderate effect against both Gram-positive and Gram-negative strains compared to the reference ciprofloxacin. Additionally, studies on the thrombolytic and hemolytic activity of the compounds were conducted [70].

Among more complex molecules, Rezki et al. (2015) received a series of symmetrically 2,5-disubsituted-thiadiazoles derivatives hybridized with a triazole, thiadiazole or oxadiazole. Compound **37** (Figure 14) with the free thione group at position 2 of 1,3,4-oxadiazole, showed the strongest antibacterial activity against *P. aeruginosa*, *S. pneumoniae* and *S. aureus* in comparison to the reference drug ciprofloxacin. Additionally, some of the newly synthesized structures possess anti-proliferative potential [71].

Moreover, researchers from India led by Neeraja (2016) decided to modify known non-steroidal anti-inflammatory drugs and combine their core with heterocycles (mercapto-1,3,4-oxadiazole and triazole). The combination of naproxen **38** (Figure 14) was the most active. The resulting hybrid compound turned out to be moderately antibacterial compared to the reference drug-amikacin [72].

In 2017, the team of Wang and Xie synthesized a series of 5-phenyl-1,3,4-oxadiazole-2-thiol substituted *N*-hydroxyethyl quaternary ammonium salts (general structure **39** in Figure 15) and investigated the structure, activity and antimicrobial effects. It turned out that the length of the carbon chain is of great importance, and the most active derivatives had a dodecyl alkyl chain. Additionally, the dependence of substituents on the phenyl group was checked. The best derivative turned out to be *tert*-butyl, which showed moderate-to-good antibacterial and antifungal activity compared to benzalkonium chloride (BZK) and chlorhexidine acetate (CA) as positive controls. Researchers also tried to elucidate the possible mechanism of action, which could be puncturing the bacterial cell membrane and releasing the cytoplasm from the cell [73,74].

Additional studies by Ambhore et al. from 2019 proved that a series of pyridin-4-yl-1,3,4-oxadiazol-2-yl-thio-ethylidene-hydrazinecarbothioamide derivatives possess antitubercular activity. The most active derivatives **40a**–**c** (Figure 15) showed MIC = 3.90 µg/mL on selected *Mycobacterium* strains, which was weaker in comparison to reference drugs: isoniazid (MIC = 0.48 µg/mL) and rifampicin (MIC = 0.24 µg/mL). Additionally, antioxidant tests and molecular docking studies were performed. These showed that the mechanism of action of the compounds may be related to the inhibition of the sterol 4α-demethylase (CYP51 enzyme). It participates in the synthesis of sterols by removing the 14α-methyl group from the sterol nucleus [75].

Furthermore, studies by Hofny et al. from 2021 prove that *S*-substituted 2-mercapto-1,3,4-oxadiazole-quinoline hybrids possess antibacterial activity. The most active derivatives **41a**, **41b** (Figure 15) showed stronger or comparable activity against Gram-negative bacteria *P. aeruginosa* and *E. coli* as well as Gram-positive *S. aureus* strains compared to the reference drug ciprofloxacin. In addition, it turned out that they are strong inhibitors of bacterial topoisomerases: II (DNA gyrase) and IV, which play the necessary functions in the process of bacterial DNA replication. Molecular docking studies confirmed the good ability to bind to the key amino acids of enzymes [37].

Also in 2021, Al-Wahaibi published the results of research on antibacterial activity of the *N*-Mannich base of 5-(3,4-dimethoxyphenyl)-1,3,4-oxadiazole-2-thione derivatives. Comparing the diameter of the inhibition zone of the new compounds and reference drugs (gentamicin, ampicillin), the synthesized derivatives showed comparable or higher inhibitory activity against bacterial growth. Compounds **42a**–**b** (Figure 15) had a broad spectrum of activity, affecting Gram-positive bacteria *S. aureus*, *B. subtilis* and *M. luteus*, as well as Gram-negative *P. aeruginosa* and *E. coli*. Compounds **42c**–**d** (Figure 15), in turn, were only active against Gram-positive strains. Additionally, some of the derivatives displayed anti-proliferative activity [76].

Among described derivatives some conclusions regarding the relationship structure-activity can be seen. In simple 5-aryl-1,3,4-oxadiazole-2-thioles structures, the substituents on the phenyl ring influence the antimicrobial activity. Antitubercular activity increases in the order 4-Cl < 4-NO_2_ < 4-H < 4-Br < 4-I < 4-OH. In turn, the activity against *P. aeruginosa* decreases in the order 4-NO_2_ > 4-CH_3_ ≥ 4-Cl ≥ 4-F ≥ 4-H ≥ 4-OH ≥ 4-N(CH_3_)_2_. Ammonium salts containing the 2-mercapto-1,3,4-oxadiazole structure showed the greatest activity when the distance between the heterocycle and the cation was 12 carbon atoms and the *tert*-butyl group was the most favored substituent on the phenyl ring. The halogen in the aromatic ring in the derivatives of *S*-substituted 2-mercapto-1,3,4-oxadiazoles enhances the antibacterial activity. Derivatives containing a 3,5-dinitrophenyl or 3,5-dinitrobenzyl-mercapto group showed the greatest activity by substituting the additional aryl ring with halogens, especially chlorine. Among the pyridine derivatives of *S*-substituted 1,3,4-oxadiazole-2-thiols, the compounds with bromine, methyl or nitro groups in the phenyl substituent showed the highest tuberculostatic activity.

## 3. Antifungal Activity of 1,3,4-Oxadiazole Derivatives

1,3,4-oxadiazole derivatives, in addition to their antibacterial activity, also had an effect on various types of fungi.

In 2015, Gavarkar and Somani designed and synthesized a series of 2,5-disubstituted 1,3,4-oxadiazoles and checked their antifungal activity. Compound **43** (Figure 16) turned out to be the most active. As compared to fluconazole, it showed 8 to 16 times greater activity against *A. niger* and *C. albicans*, respectively. Additionally, several derivatives displayed an antimycobacterial effect [77].

In addition, Wani and his team (2015) developed 1,3,4-oxadiazole derivatives bearing pyridine and imidazole scaffolds. The most active compounds **44a**–**c** (Figure 16) showed a stronger effect than fluconazole against several *Candida* strains. The suggested mechanism has been confirmed in molecular docking studies and may concern inhibition of ergosterol synthesis, as the new structures fit very well in the active site of the lanosterol-14α-demethylase enzyme. It takes part in the synthesis of ergosterol, the main sterol component of the fungal cell membrane. Inhibition of this enzyme causes loss of cell continuity and cell dysfunction [78].

Moreover, researchers from China under the direction of Liao (2015) obtained 1,2,3-triazolo-analogues of fluconazole containing an additional 1,3,4-oxadiazole ring. The most active derivatives **45a** and **45b** (Figure 16) showed excellent broad-spectrum antifungal activity (MICs_80_ ≤ 0.125 µg/mL) up to 64 times higher than the reference itraconazole and fluconazole (MICs_80_ = 0.5–8 µg/mL). The possible mechanism of action of the compound **45a** was confirmed by molecular docking studies on the active side of cytochrome P450 14α-demethylase (CYP51) [79].

Nimbalkar et al. (2016) obtained a number of Mannich bases containing 1,3,4-oxadiazole-2-thione in the structure and examined their antifungal potential. The most active derivatives **46** and **47a**–**b** (Figure 17) showed stronger effect than the standard drug fluconazole against *A. fumigatus* and *C. glabrata* strains. Additionally, during docking studies, new compounds were characterized by good binding affinity to the active side of lanosterol-14α-demethylase enzyme [80].

In addition, researchers from Romania under the direction of Stoica (2016) synthesized a series of 1,3,4-oxadiazole derivatives containing a thiazole ring in their structure and tested in silico their activity against the lanosterol 14α-demethylase enzyme from three fungal strains. The results of the molecular docking study showed that the most active derivatives were **48a**–**b** (Figure 17), which inhibited the enzyme obtained from *Aspergillus fumigatus*, comparable to the reference ketoconazole and voriconazole [81].

The Frost–Revie team (2016, 2020) first obtained peptide macrocycles containing a 1,3,4-oxadiazole ring **49a-b** (Figure 18) and then tested their antifungal activity. It turned out that some structures with an 18-membered ring showed themselves active against *Candida albicans*, but additionally, they act synergistically to the antifungal drug-fluconazole, increasing its effectiveness. Combination therapy creates new opportunities to deal with microorganisms that are resistant to drugs already in use [82,83].

Another research was conducted in 2017 by Levent et al. on 1,3,4-oxadiazole-1,3,4-thiadiazole derivatives and their antifungal activity. The derivatives containing the nitro substituent **50a**–**c** (Figure 18), the introduction of which enhanced the effect, turned out to be the most active. The new structures displayed a strong activity (MICs_50_ = 0.78–3.12 µg/mL) against four *Candida* strains compared to ketoconazole (MICs_50_ = 0.78–1.56 µg/mL). Additional docking studies showed a probable mechanism of action as compound **50a** bound strongly to the active side of lanosterol-14α-demethylase enzyme [84].

In 2019, researchers from Turkey led by Karaburun (2019) published the results of the antifungal potential of new synthesized 1,3,4-oxadiazole-benzimidazole hybrids. The most active derivatives **51a**–**b** (Figure 18) were as strongly active against the *C. albicans* strain as the reference amphotericin B, and almost 4 times stronger than ketoconazole. Additionally, molecular docking studies were performed, which may suggest that their mechanism of action is related to the inhibition of ergosterol synthesis [85].

Moreover, Bordei Telehoiu et al. (2020) designed and synthesized a number of compounds containing 6-chloro-9*H*-carbazole; some of them additionally possess a 1,3,4-oxadiazole ring. They were tested for antibacterial and antifungal activity. The most active derivative turned out to be the compound **52** (Figure 18), which showed by far the strongest activity against the *Candida albicans* strain [86].

When analyzing the data presented above, in antifungal structures, it can be seen that one substituent is preferred over the other and also the site of substitution is crucial. Electron-withdrawing groups, e.g., NO_2_, play an important role, which additionally affects the physicochemical properties of the compound (lipophilicity, hydrophobic interactions, and pKa). Some derivatives are synthesized on the basis of already known drugs (fluconazole), and other structures, in addition to their own activity, may increase the effectiveness of standard antifungal drugs.

## 4. Antiprotozoal Activity of 1,3,4-Oxadiazole Derivatives

Among the antiprotozoal derivatives of 1,3,4-oxadiazole we can distinguish structures that affect specific species, e.g., *Plasmodium* spp., *Trypanosoma* spp., and *Leishmania* spp.

### 4.1. Antimalarial Activity of 1,3,4-Oxadiazole Derivatives

In 2016, Radini et al. obtained heterocyclic derivatives with quinolinyl residue. One of the most potent antimalarial compounds was the derivative containing the 1,3,4-oxadiazole ring **53** (Figure 19). It exhibited similar half-maximal inhibitory concentration (IC_50_) against *Plasmodium falciparum* as compared to the reference drug chloroquine [87].

Additionally, the antimalarial activity of the synthesized 5-mercapto-1,3,4-oxadiazole derivatives was investigated by Thakkar et al. (2017). Among the obtained Schiff bases, compound **54** (Figure 19) proved to be the strongest, showing three times higher activity than the reference pyrimethamine. Additionally, the authors suggested the mechanism of action of the new derivatives as inhibitors of dihydrofolate reductase (DHFR) following in vitro DHFR enzyme inhibition assay. DHFR participates in the synthesis, for example purine bases, therefore, inhibition of this enzyme results in impaired proliferation and cell growth. New derivatives also showed antibacterial activity [88].

Moreover, the Verma team (2018–2019) designed and synthesized several series of pyrazole-vinyl-1,3,4-oxadiazole hybrids derivatives. The most active compounds were **55a** with an additional indole ring and **55b** bearing furan moiety (Figure 19). Antiparasitic tests were performed on chloroquine-sensitive *Plasmodium falciparum* 3D7. Compared to the standard drug chloroquine (IC_50_ = 0.402 µg/mL) the structures **55a**–**b** displayed enhanced activity (IC_50_ = 0.245/0.248 µg/mL). Additionally, the structures strongly inhibited the enzyme falcipain-2 (IC_50_ = 7 or 14 µM), and molecular docking studies confirmed good affinity to that enzyme [89,90]. Falcipain-2, a cysteine protease from *Plasmodium falciparum*, is a key enzyme that plays an important role in the degradation of hemoglobin in protozoa. Blocking this enzyme causes the accumulation of undigested hemoglobin and inhibition of parasite development, and is therefore an important target of antimalarial drugs [91]. Some new derivatives also showed antileishmanial activity.

### 4.2. Antileishmanial Activity of 1,3,4-Oxadiazole Derivatives

Researches under the direction of Taha (2017) synthesized two series of 1,3,4-oxadiazole derivatives and checked their antileishmanial activity. Among the compounds of the first series containing benzohydrazone scaffold, the most active were the derivatives with di- or trihydroxyphenyl substituent **56a** and **56b** (Figure 20), which were about 7 times more potent (IC_50_ = 0.95/0.98 µM) than the reference pentamidine (IC_50_ = 7.02 µM). On the other hand, in the second series of quinolinyl-1,3,4-oxadiazole linked to a thiosemicarbazide via a phenyl linker, the strongest derivatives **57a**–**b** (Figure 20) showed even 70 times greater activity than the same reference drug (IC_50_ = 0.10/0.15 µM). Molecular docking studies suggest a potential mechanism of action of the newly formed compounds as pteridine reductase (PTR1) inhibitors. This enzyme is involved in pteridine salvage, therefore, its inhibition disrupts the function of the protozoan cell [92,93].

Furthermore, researchers from Brazil led by Chaves and Espinosa (2017, 2020) received a series of gold(I) phosphine complexes with 2-mercapto-1,3,4-oxadiazoles and checked their activity against *Leishmania spp.* From the series of phenyl derivatives, the structure **58** (Figure 20) with triethylophosphine complex turned out to be the most active against *L. infantum* intracellular amastigotes. In turn, the most active four compounds **59a**–**b**, **60a**–**b** (Figure 20) derived from δ-*D*-gluconolactone showed the ability to inhibit the proliferation of *L. braziliensis* intracellular amastigotes. This form of the parasite is an important target in therapy. Additionally, the compounds displayed anticancer activity [94,95].

### 4.3. Antitrypanocidal Activity of 1,3,4-Oxadiazole Derivatives

Patel’s team (2017) obtained a number of 1,3,4-oxadiazole-quinoxaline derivatives and extensively investigated their antimicrobial activity. Antiprotozoal activity was only directed against *Trypanosoma cruzi*. The strongest compounds **61a**–**b** (Figure 21), showed similar or better effect against *Trypanosoma cruzi* trypomastigotes in comparison to the standard drugs benznidazole and nifurtimox. Additionally, some compounds showed good antibacterial activity, while others showed antifungal activity [96].

In addition, Shaykoon et al. (2020) synthesized a series of pyrazine-based derivatives of 1,3,4-oxadiazole or 1,2,4-triazole and examined their antitrypanocidal activity. Triazole derivatives showed greater activity, but among the compounds with 1,3,4-oxadiazole, the most active derivative against *T. brucei* was **62** (Figure 21) (IC_50_ = 3.94 µM). It was stronger than α-difluoromethylornithine (DFMO) which was used as a control (IC_50_ = 6.1 µM) [97].

In 2021, Ribeiro et al. developed a series of 1,3,4-oxadiazole bearing pyrazolo[3,4-*b*]pyridine scaffold derivatives and tested their antitrypanocidal activity. The most potent compound in vitro was **63** (Figure 21), whose activity against amastigotes forms of *T. cruzi* (IC_50_ = 1.11 µM) was significantly higher than that of the reference drug benznidazole (IC_50_ = 3.98 µM) [98].

When discussing structure-activity relationship of antiprotozoal compounds, it can be stated that often an additional heterocyclic ring in direct connection with 1,3,4-oxadiazole significantly increases the activity the whole molecule. Additionally, strong electron withdrawing substituents, e.g., CF_3_, are preferred to enhance the activity. In the case of halogens, the activity decreases in the order F > Cl > Br.

## 5. Antiviral Activity of 1,3,4-Oxadiazole Derivatives

Taking into account the broadly understood concept of antimicrobial effect, 1,3,4-oxadiazole derivatives also showed antiviral activity.

Researchers from France led by Benmansour (2016) examined the antiviral activity of the new 1,3,4-oxadiazole derivatives bearing a thiofen nucleus. They investigated the effectiveness of the new compounds as dengue virus (DENV) inhibitors targeting NS5 (nonstructural protein 5) polymerase, which is an essential RNA-dependent RNA polymerase (RdRp) that is strictly required for the replication of viruses. The most active derivatives were **64a**–**d** (Figure 22) whose IC_50_ on DENV-2 RdRp activity was ≤6 µM. In addition, their activity was also tested on four different clinically isolated serotypes (DENV1-4) and compound **64d** proved to be the most effective [99].

In 2018, Tawfik and his team designed and synthesized a number of 1,3,4-oxadiazole derivatives and determined their activity against the influenza virus on the basis of plaque reduction assay. Amantadine was used as the reference drug, which showed 100% effectiveness in inhibiting the multiplication of the influenza virus. Some of the new compounds were highly active, but only the 2-(4-chloro-3-nitrophenyl)-5-(*p*-tolyl)-1,3,4-oxadiazole derivative **65** (Figure 22) was as effective as the standard drug. Additionally, the IC_50_ value (39 µg/µL) confirmed its strongest antiviral activity [100].

Moreover, Albratty et al. (2019) obtained a number of 1,3,4-oxadiazole derivatives and checked their antiviral activity. Compounds **66a** and **66b** (Figure 22) additionally bearing an aminothiazole substituent were most active against *Feline herpes virus* (FHV), *Feline corona virus* (FCoV), *Herpes simplex virus-1* (HSV-1 KOS) and *Herpes simplex virus-2* (G). Their values of IC_50_ for the *Herpes* strains were 2 µmol/L, so they were weaker compared to acyclovir IC_50_ = 0.4 µmol/L, but stronger than cydofovir IC_50_ = 125–250 µmol/L. In turn, compound **67** (Figure 22) containing the phenyl hydrazonoyl group was most active against *Vaccinia virus*, *Herpes simplex virus-1* (TK-KOS-ACVr—thymidine kinase-deficient HSV-1 KOS strain resistant to acyclovir), *Vesicular stomatitis virus* and *Coxsackie virus B4*. The IC_50_ value of the compound was 4 µmol/L and it was lower than the reference acyclovir IC_50_ = 50–250 µmol/L and the cydofovir IC_50_ = 10–250 µmol/L, so the new derivative was more potent. Some compounds also demonstrated cytotoxic activity [101].

In addition, El Mansouri et al. (2020) focused their research on assessing the antiviral activity of synthesized homonucleoside analogues containing a 1,3,4-oxadiazole ring. All the compounds were checked for activity against the human varicella-zoster virus (VZV), both wild-type and thymidine kinase-deficient (TK-). 6-Azauracil derivative **68** (Figure 22) showed twice the activity compared to aciclovir, which was used as a control against VZV TK-. The effective concentration required to reduce the virus plaque formation by 50% (EC_50_) of **68** was 49.67 µM with an EC_50_ = 103.23 µM for the reference medicine. Due to the fact that the VZV TK- strain is resistant to conventional antiviral drugs, the synthesized derivative can serve as a leading structure for further research. Additionally, some of the derivatives showed cytotoxic activity [102].

In 2020, researchers from Pakistan under the direction of Hamdani developed a series of 1,3,4-oxadiazole-benzenesulfonamide-*S*-alkylphthalimide hybrids. They examined the efficacy of the obtained compounds against dengue virus (DENV) by checking their activity against dengue protease consisting of two nonstructural proteins NS2B/NS3pro. The enzyme has proteolytic properties and also influences viral replication. Among the synthesized 1,3,4-oxadiazole derivatives, the compounds **69a** and **69b** (Figure 22) were some of the best. Their IC_50_ were 13.9 µM and 15.1 µM, respectively. Moreover, the **69a** derivative showed the highest percent (99.9%) inhibition of DENV2 NS2B/NS3 protease activity. Molecular docking studies confirmed the good binding affinity to the active site of that enzyme [103].

In summary it can be emphasized that the activity of antiviral compounds is higher when an electron withdrawing substituent, e.g., trifluoromethyl or halogens, appears in the structure.

## 6. Summary

The conducted review of 1,3,4-oxadiazole derivatives shows a high potential of their antimicrobial activity. The new structures have a very wide spectrum of activity, from bacteria and fungi to protozoa and viruses. Probable mechanisms of action of some of the new compounds are described, which are based on the inhibition of various enzymes, e.g., DNA gyrase, enoyl reductase, and lanosterol-14α-demethylase. Many of the new derivatives exceed the activity of the already known antimicrobial drugs. The variety of new structures and their high activity confirm their value as new drugs in the fight against antimicrobial resistance. Further studies are necessary to confirm their effectiveness and safety in vivo.

## Figures and Tables

**Figure 1 ijms-22-06979-f001:**
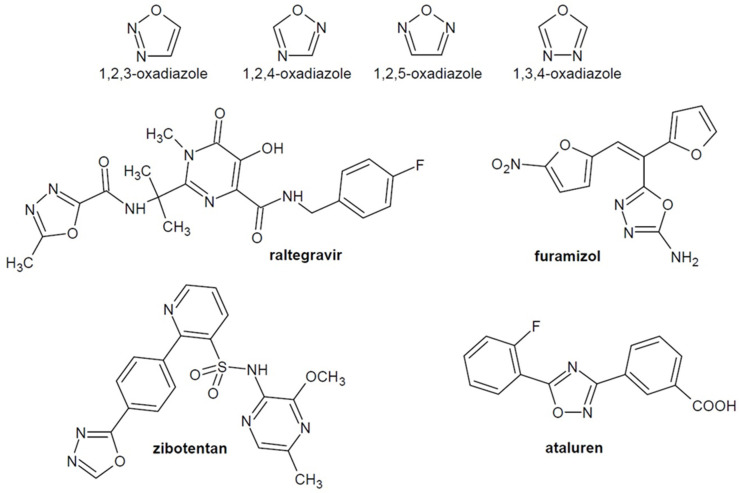
Isomeric forms of oxadiazole and examples of known drugs containing an oxadiazole ring.

**Figure 2 ijms-22-06979-f002:**
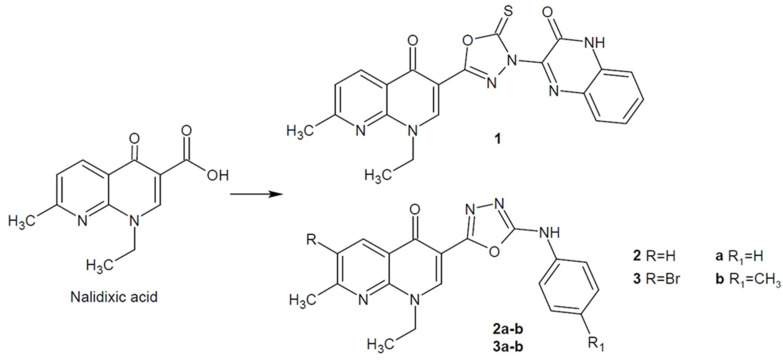
1,3,4-oxadiazole hybrids of nalidixic acid with antibacterial activity.

**Figure 3 ijms-22-06979-f003:**
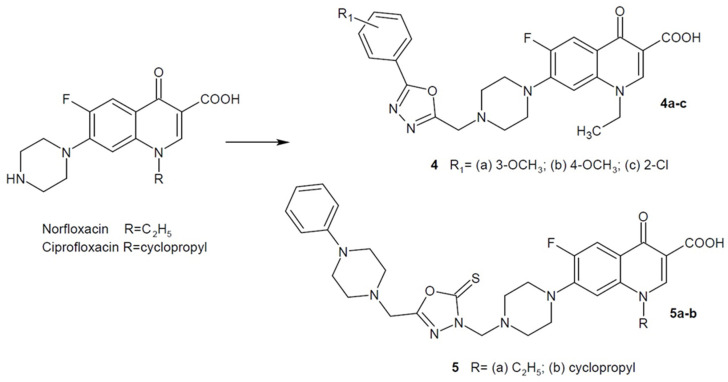
1,3,4-oxadiazole hybrids of fluoroquinolone with antibacterial activity.

**Figure 4 ijms-22-06979-f004:**
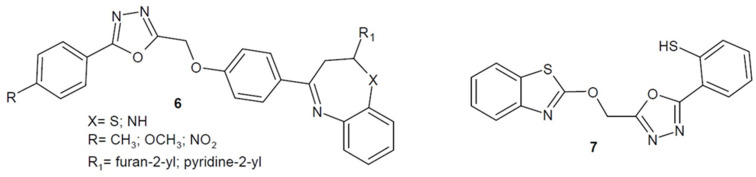
Aryl-1,3,4-oxadiazole derivatives with antibacterial activity.

**Figure 5 ijms-22-06979-f005:**
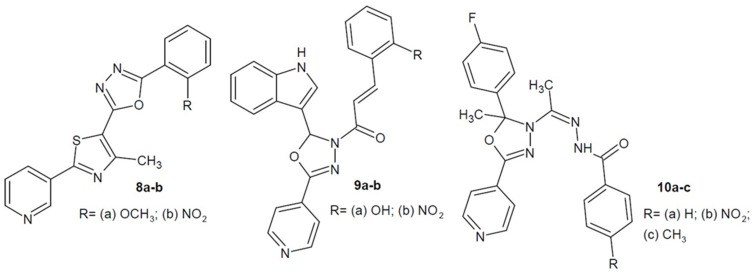
Aryl and/or heteroaryl 1,3,4-oxadiazole derivatives with antibacterial activity (part 1).

**Figure 6 ijms-22-06979-f006:**
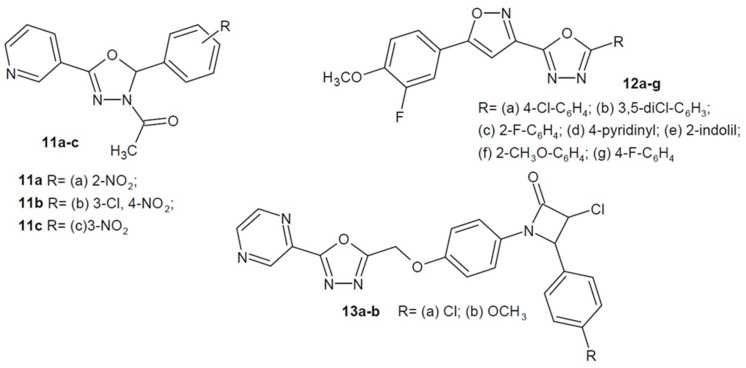
Aryl and/or heteroaryl 1,3,4-oxadiazole derivatives with antibacterial activity (part 2).

**Figure 7 ijms-22-06979-f007:**
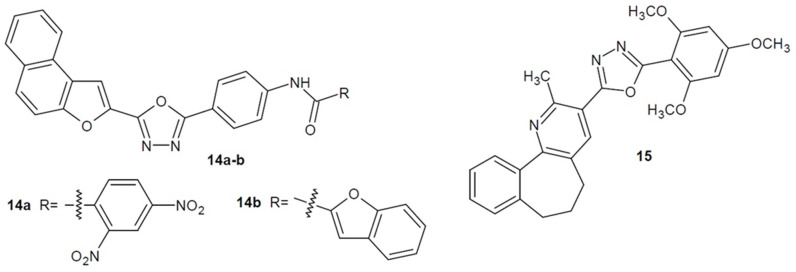
Aryl and/or heteroaryl 1,3,4-oxadiazole derivatives with antibacterial activity (part 3).

**Figure 8 ijms-22-06979-f008:**
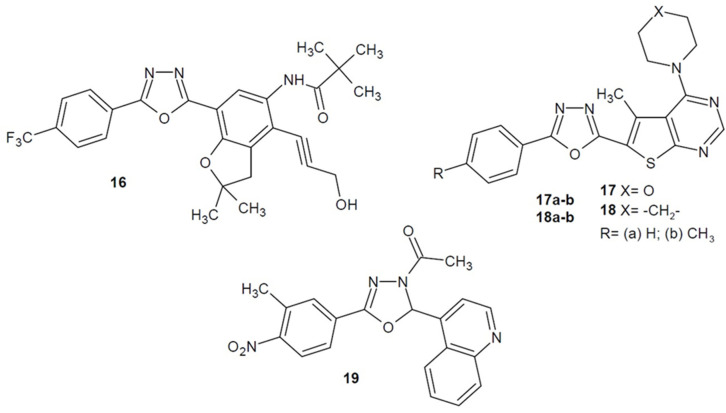
Aryl and/or heteroaryl 1,3,4-oxadiazole derivatives with antibacterial activity (part 4).

**Figure 9 ijms-22-06979-f009:**
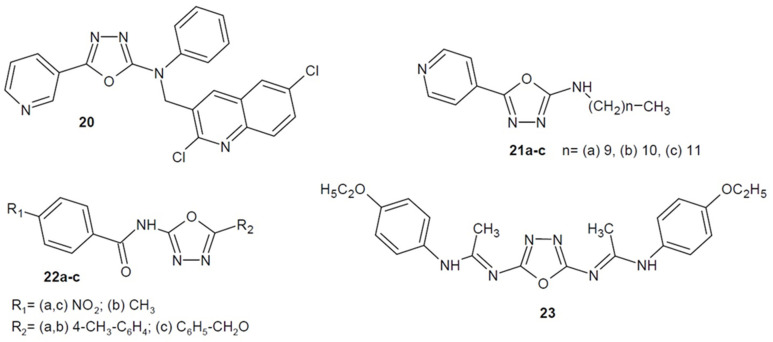
Amino-1,3,4-oxadiazole derivatives with antibacterial activity (part 1).

**Figure 10 ijms-22-06979-f010:**
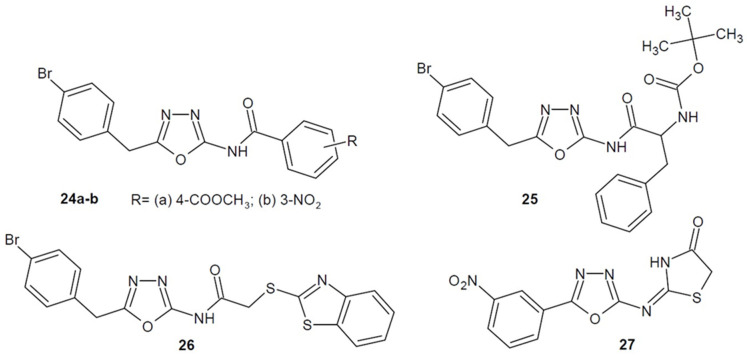
Amino-1,3,4-oxadiazole derivatives with antibacterial activity (part 2).

**Figure 11 ijms-22-06979-f011:**
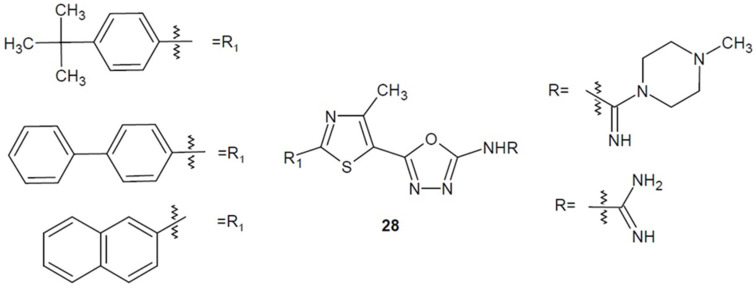
Amino-1,3,4-oxadiazole derivatives with antibacterial activity (part 3).

**Figure 12 ijms-22-06979-f012:**
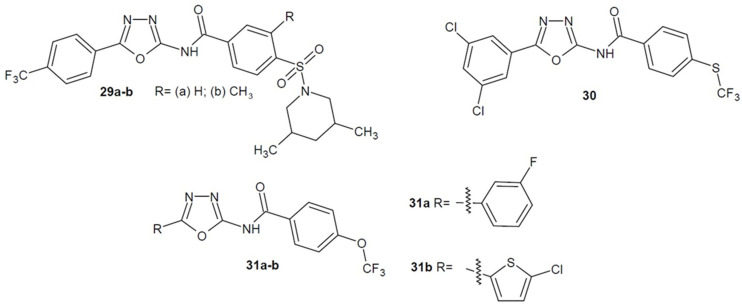
Amino-1,3,4-oxadiazole derivatives with antibacterial activity (part 4).

**Figure 13 ijms-22-06979-f013:**
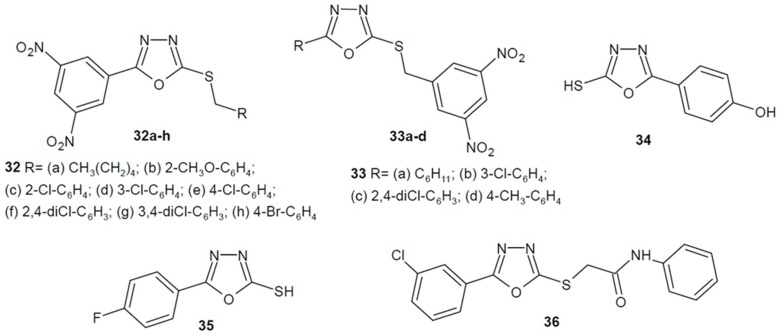
Simple 1,3,4-oxadiazole-2-thiols and their *S*-substituted derivatives with antibacterial activity.

**Figure 14 ijms-22-06979-f014:**
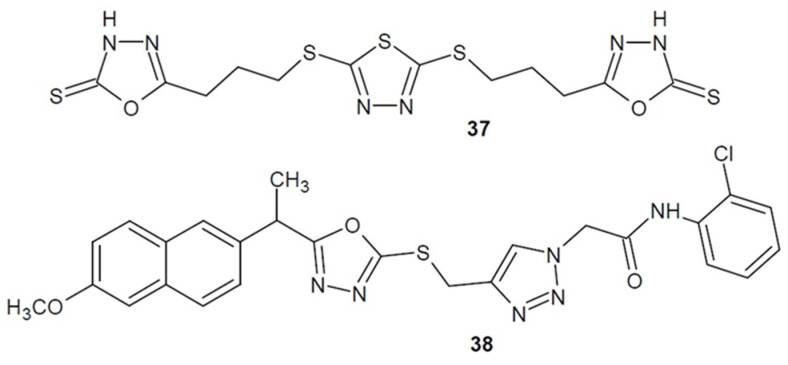
1,3,4-oxadiazole-2-thiones/thiols and their *S*-substituted derivatives with antibacterial activity (part 1).

**Figure 15 ijms-22-06979-f015:**
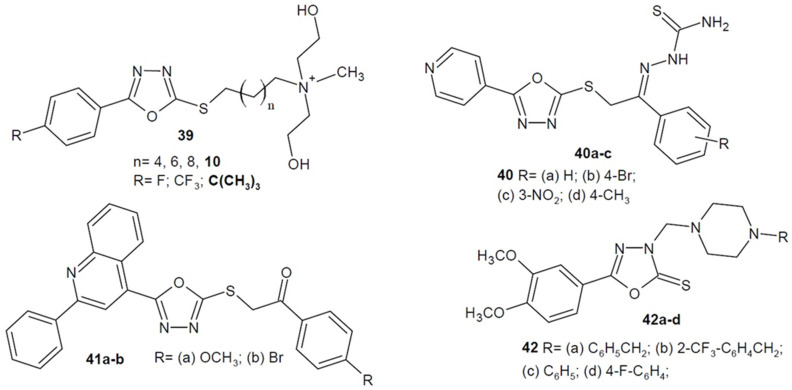
1,3,4-oxadiazole-2-thiones/thiols and their *S*-substituted derivatives with antibacterial activity (part 2).

**Figure 16 ijms-22-06979-f016:**
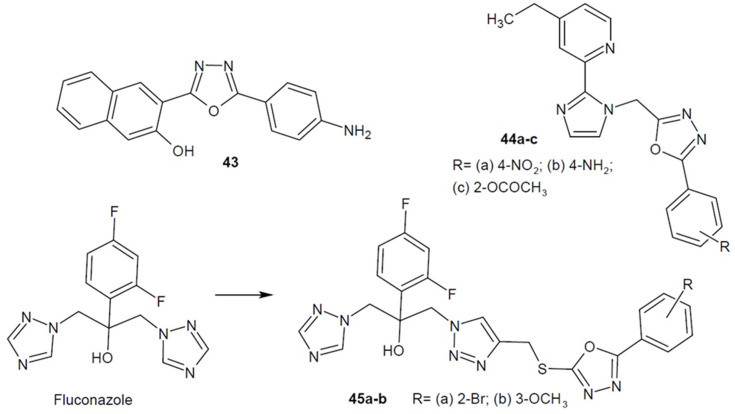
1,3,4-oxadiazole derivatives with antifungal activity (part 1).

**Figure 17 ijms-22-06979-f017:**
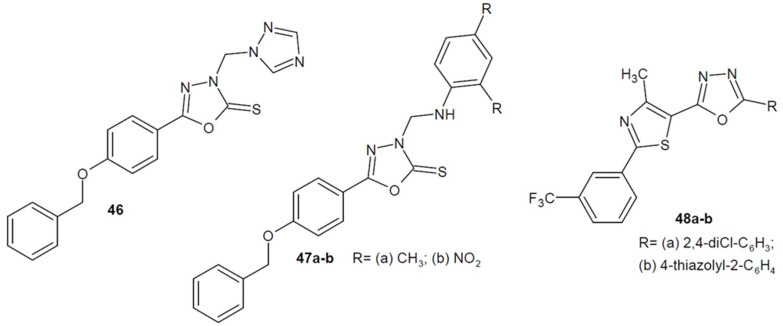
1,3,4-oxadiazole derivatives with antifungal activity (part 2).

**Figure 18 ijms-22-06979-f018:**
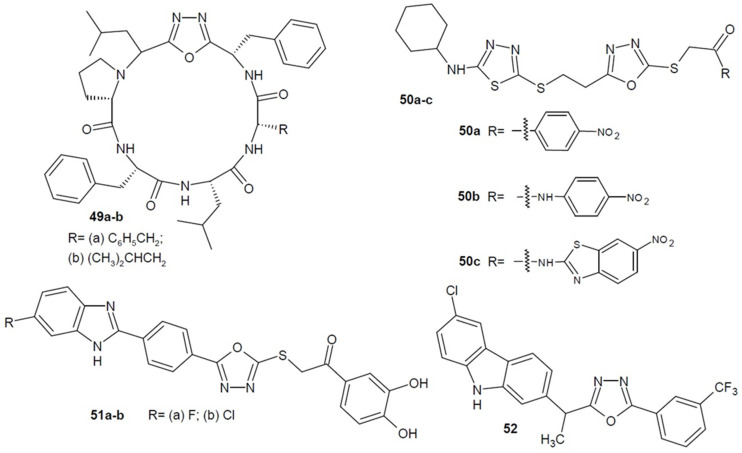
1,3,4-oxadiazole derivatives with antifungal activity (part 3).

**Figure 19 ijms-22-06979-f019:**
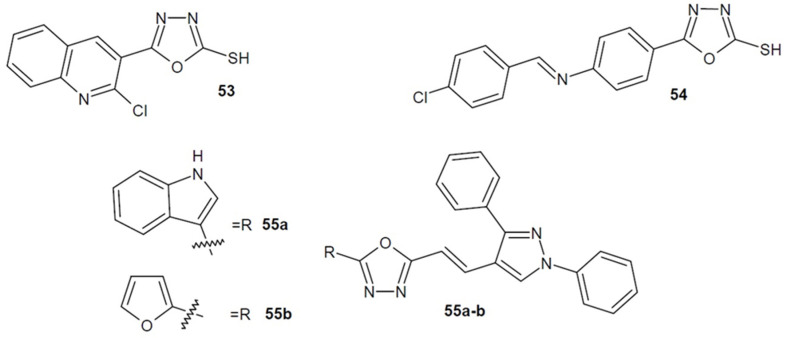
1,3,4-oxadiazole derivatives with antimalarial activity.

**Figure 20 ijms-22-06979-f020:**
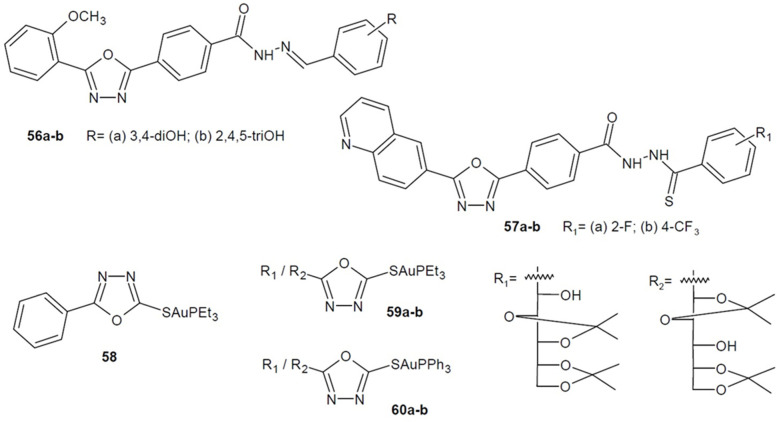
1,3,4-oxadiazole derivatives with antileishmanial activity.

**Figure 21 ijms-22-06979-f021:**
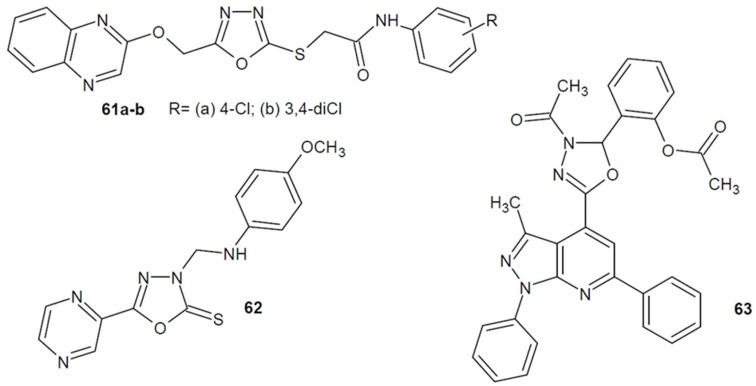
1,3,4-oxadiazole derivatives with antitrypanocidal activity.

**Figure 22 ijms-22-06979-f022:**
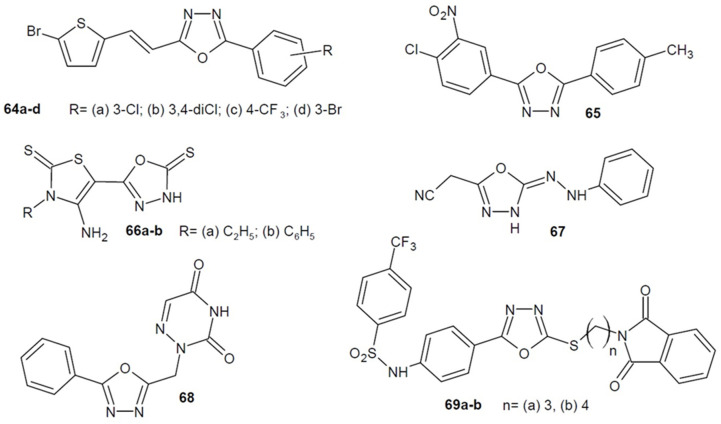
1,3,4-oxadiazole derivatives with antiviral activity.

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
