# Peer review of "Antimicrobial Activity of 1,3,4-Oxadiazole Derivatives"

_ijms, 2021, doi:10.3390/ijms22136979_

Round 1

Reviewer 1 Report

Just have some minor English mistakes 

Author Response

Dear Reviewer,

We sincerely appreciate your reviewing our contribution and giving us another opportunity to improve our manuscript in the best conceivable way. We considered all comments as constructive criticism. Considering the Reviewer comments, we have made a revision of our manuscript and our detailed answers are included below.

Just have some minor English mistakes

The manuscript has been checked and corrected.

Reviewer 2 Report

Dear Authors

I had the honor to look through your review. I can say that it is a good review, detailed, comprehensive and contains adequate references for a review. Current work summarizes the research on old and newly synthesized 1,3,4-oxadiazole derivatives for their antimicrobial activity. 

The abstract, introduction and summary are according to requirements, so do the references. 

I have only very minor comments: 

  1. At the end of each section, there is a concluding summary, I feel that more can be elaborated on the "Antiviral" , "Antiprotozoal" and "Antifungal" activities.
  2. There are some English mistakes that should be corrected, I will show some of them:  line 10 in the Abstract, "which showed" should be "have shown". Line 118: remove the word "the" before "antitubecular". Replace the word "on Figure..." to "in figure ... as in lines 267, 349, ... Please be consistent in writing the names of compounds, example "mercapto" not "merkapto" as in line 342. Also, in line 405, the word "design" should be "designed". In line, the order contains "4-J" I think it should be 4-I. 

Author Response

Dear Reviewer,

We sincerely appreciate your reviewing our contribution and giving us another opportunity to improve our manuscript in the best conceivable way. We considered all comments as constructive criticism. Considering the Reviewer comments, we have made a revision of our manuscript and our detailed answers are included below.

  1. At the end of each section, there is a concluding summary, I feel that more can be elaborated on the "Antiviral" , "Antiprotozoal" and "Antifungal" activities.

In the summary of each section, there are informations that results from the analysis of the relationship between the activity of compounds and their structure. If there are too few of them, it means that no more dependencies could be found.

  1. There are some English mistakes that should be corrected, I will show some of them: line 10 in the Abstract, "which showed" should be "have shown". Line 118: remove the word "the" before "antitubecular". Replace the word "on Figure..." to "in figure ... as in lines 267, 349, ... Please be consistent in writing the names of compounds, example "mercapto" not "merkapto" as in line 342. Also, in line 405, the word "design" should be "designed". In line, the order contains "4-J" I think it should be 4-I.

All mistakes have been corrected.

Reviewer 3 Report

Manuscript ID: ijms-1264220

Type of manuscript: Review

Title: Antimicrobial Activity of 1,3,4-Oxadiazole Derivatives

Authors: Teresa Glomb, Piotr ÅšwiÄ…tek *

Submitted to section: Molecular Biophysics

This review summarized “Antimicrobial Activity of 1,3,4-Oxadiazole Derivatives” based on the literatures from 2015-2021. This chemistry is very important in the current medical chemistry. However, I found many questionable points, therefore it should be major revisions to the next step.

Overall

The redundancy expressions were found such as “a lot of” and “more and more”, They should replace. And of course, the English checking should need to the next step.

I think “the literature” should be “the literatures” in the last line of “Abstract”.

  1. Introduction

The citations of [1–6] are too ambiguous. You should cite the appropriate positions.

This review focuses on the literatures from 2015 to 2021. However, I think many compounds were reported. So, you should summarize the history of the 1,3,4-oxadiazole derivatives, as mentioned below again. Only half page of page 2 is too small.

  1. Antibacterial Activity of 1,3,4-Oxadiazole Derivatives and others

Some sentences noted the name of the country such as “Researchers from Saudi Arabia” line 120 (page 4). It was too strange. The name of the country is important? The author’s name is more important!!

2.4. Antibacterial Activity of 1,3,4-Oxadiazole-2-Thiones/Thiols and their S-Substituted Derivatives

In lines 389 and 390, these expressions are difficult to understand because the inequality sign is reversed such as “increases in the order 4-Cl<4-NO2<4-H<4-Br<4-J<4-OH.” and “decreases in order 4-NO2>4-CH3≥4-Cl≥4-F≥4-H≥4-OH≥4-N(CH3)2”.

And what is “4-J”?

And “≥” is correct? It means to include the equal ?

  1. Antiprotozoal Activity of 1,3,4-Oxadiazole Derivatives

Before 4.1 section, you should summarize what type of antibacterial activities are present.

References

The general introduction should be “OLD” compared with the years from 2015 to 2021. However, you only cited after 2015 review in ref. 1-6. I think these antimicrobial activity of 1,3,4-oxidiazole derivatives are well known chemistry. You should summarize this chemistry before your summary from 2015 to 2021.

Author Response

Dear Reviewer,

We sincerely appreciate your reviewing our contribution and giving us another opportunity to improve our manuscript in the best conceivable way. We considered all comments as constructive criticism. Considering the Reviewer comments, we have made a revision of our manuscript and our detailed answers are included below.

Overall

The redundancy expressions were found such as “a lot of” and “more and more”, They should replace. And of course, the English checking should need to the next step. I think “the literature” should be “the literatures” in the last line of “Abstract”.

The manuscript has been checked and corrected. All mistakes were corrected according to the Reviewer's suggestions.

Introduction

The citations of [1–6] are too ambiguous. You should cite the appropriate positions.

The citations were separated and assigned to the information appearing in the manuscript.

This review focuses on the literatures from 2015 to 2021. However, I think many compounds were reported. So, you should summarize the history of the 1,3,4-oxadiazole derivatives, as mentioned below again. Only half page of page 2 is too small.

The introduction has been changed. Informations on previous studies of oxadiazole derivatives has been added.

Antibacterial Activity of 1,3,4-Oxadiazole Derivatives and others

Some sentences noted the name of the country such as “Researchers from Saudi Arabia” line 120 (page 4). It was too strange. The name of the country is important? The author’s name is more important!!

The names of the authors have been added throughout the manuscript as suggested by the Reviewer.

Antibacterial Activity of 1,3,4-Oxadiazole-2-Thiones/Thiols and their S-Substituted Derivatives

In lines 389 and 390, these expressions are difficult to understand because the inequality sign is reversed such as “increases in the order 4-Cl<4-NO2<4-H<4-Br<4-J<4-OH.” and “decreases in order 4-NO2>4-CH3≥4-Cl≥4-F≥4-H≥4-OH≥4-N(CH3)2”.

This notation suggests that the compound with the 4-Cl substituent has the weakest activity, and that the compound with the 4-OH substituent turned out to be the strongest. Similarly, in the next line, the compound with the 4-NO2 substituent is the most active and then the activity decreases.

And what is “4-J”?

4-I of course

And “≥” is correct? It means to include the equal ?

Yes, that is correct. This means that the activity of the compounds was higher than or equal to.

Antiprotozoal Activity of 1,3,4-Oxadiazole Derivatives

Before 4.1 section, you should summarize what type of antibacterial activities are present.

Before section 4.1 there is the following sentence: “Among the antiprotozoal derivatives of 1,3,4-oxadiazole we can distinguish structures that affect specific species, e.g. Plasmodium spp., Trypanosoma spp., Leishmania spp.

References

The general introduction should be “OLD” compared with the years from 2015 to 2021. However, you only cited after 2015 review in ref. 1-6. I think these antimicrobial activity of 1,3,4-oxidiazole derivatives are well known chemistry. You should summarize this chemistry before your summary from 2015 to 2021.

The introduction has been changed. Informations on previous studies of oxadiazole derivatives has been added.

Reviewer 4 Report

In the present review article, a survey of  1,3,4-oxadiazole derivatives and their antimicrobial activity have been presented. These compounds have shown antibacterial, antitubercular, antifungal, antiprotozoal and antiviral activity. Considering this, the content of the review is significant and interesting, nevertheless, a few issues should be corrected before its acceptance to the International Journal of Molecular Sciences.

  • MIC values are expressed in μg/mL for some compounds, while for the other in μM. My suggestion is to use both units for all presented compounds.
  • The units, such as ml or mL or cm3, should be written uniformly.
  • All abbreviations should be defined at first appearance, such as MIC or EC50.
  • Instead of 4-J, it should be 4-I.
  • Instead of gold (I), it should be gold(I).

Author Response

Dear Reviewer,

We sincerely appreciate your reviewing our contribution and giving us another opportunity to improve our manuscript in the best conceivable way. We considered all comments as constructive criticism. Considering the Reviewer comments, we have made a revision of our manuscript and our detailed answers are included below.

MIC values are expressed in μg/mL for some compounds, while for the other in μM. My suggestion is to use both units for all presented compounds.

We present the MIC values as presented in the source publications.

The units, such as ml or mL or cm3, should be written uniformly.

All values were corrected to mL.

All abbreviations should be defined at first appearance, such as MIC or EC50.

Corrected

Instead of 4-J, it should be 4-I.

Corrected

Instead of gold (I), it should be gold(I).

Corrected

Reviewer 5 Report

The current review article by Glomb and ÅšwiÄ…tek is very comprehensively written and it provides a very detailed review on the topic.

I have few concerns regarding this article.

  1. Plagiarism is 30 %. This needs to be addressed.
  2. Apart from it, minor grammatical corrections are needed.
  3. Line 242. Group B??
  4. Line 251. “an” NADH

Author Response

Dear Reviewer,

We sincerely appreciate your reviewing our contribution and giving us another opportunity to improve our manuscript in the best conceivable way. We considered all comments as constructive criticism. Considering the Reviewer comments, we have made a revision of our manuscript and our detailed answers are included below.

Plagiarism is 30 %. This needs to be addressed.

After searching the available databases, we can conclude that there is no similar review publication on the antimicrobial activity of oxadiazole derivatives in the last 5 years. All publications used in the preparation of the manuscript are cited.

Apart from it, minor grammatical corrections are needed.

The manuscript has been checked and corrected.

Line 242. Group B??

Corrected

Line 251. “an” NADH

Corrected

Round 2

Reviewer 3 Report

Manuscript ID: ijms-1264220-r1

Type of manuscript: Review

Title: Antimicrobial Activity of 1,3,4-Oxadiazole Derivatives

Authors: Teresa Glomb, Piotr ÅšwiÄ…tek *

Submitted to section: Molecular Biophysics

The English writing is not so perfect.

You should apply English Editing service to complete this manuscript editing.

References

Some points which I pointed out were revised. However, the history of 1,3,4-Oxadiazole was not noted. Almost references are after 2015. This compound was well-known, so many chemistry has been done before. It is the review article, even if you summarized the literatures from 2015-2021. From this reason, you should add the history.

Author Response

Dear Reviewer,

Thanks again for your review of our manuscript. I hope that the corrections we have made are sufficient and that our manuscript meets your requirements in the current version.

The English writing is not so perfect.

You should apply English Editing service to complete this manuscript editing.

The manuscript has been checked and corrected.

References

Some points which I pointed out were revised. However, the history of 1,3,4-Oxadiazole was not noted. Almost references are after 2015. This compound was well-known, so many chemistry has been done before. It is the review article, even if you summarized the literatures from 2015-2021. From this reason, you should add the history.

The introduction has been improved. A section that describes the discovery and history of oxadiazole derivatives has been added.

Reviewer 5 Report

All the corrections are incorporated.

Acceptable in present form. 

Author Response

Dear Reviewer,

Thank you for your acceptance of the revised manuscript.